**EMBO** *reports*

# Molecular architecture of glideosome and nuclear F-actin in *Plasmodium falciparum*

Vojtěch Pražák[1,2,3], Daven Vasishtan[1,2,3], Kay Grünewald [ID][1,2,3,4], Ross G Douglas[5,6] & Josie L Ferreira [ID][7✉]

## Abstract

**Actin-based motility is required for the transmission of malaria sporozoites. While this has been shown biochemically, filamentous actin has remained elusive and has not been directly visualised inside the parasite. Using focused ion beam milling and electron cryo-tomography, we studied dynamic actin filaments in unperturbed *Plasmodium falciparum* cells for the first time. This allowed us to dissect the assembly, path and fate of actin filaments during parasite gliding and determine a complete 3D model of F-actin within sporozoites. We observe micrometre long actin filaments, much longer than expected from in vitro studies. After their assembly at the parasite's apical end, actin filaments continue to grow as they are transported down the cell as part of the glideosome machinery, and are disassembled at the basal end in a rate-limiting step. Large pores in the IMC, constrained to the basal end, may facilitate actin exchange between the pellicular space and cytosol for recycling and maintenance of directional flow. The data also reveal striking actin bundles in the nucleus. Implications for motility and transmission are discussed.**

**Keywords** Malaria; Plasmodium; Actin; Motility; Cryo-electron Tomography
**Subject Categories** Cell Adhesion, Polarity & Cytoskeleton; Microbiology, Virology & Host Pathogen Interaction

## Introduction

The *Plasmodium falciparum* parasite causes the most severe form of malaria in humans (Cowman et al, 2016). Infection occurs during a bite from an infected mosquito, where sporozoites leave the mosquito salivary glands and are deposited into the skin (Amino et al, 2006). Within the skin, sporozoites move rapidly ($1–2\,\mu\text{m s}^{-1}$) and persistently (more than 1 h) to encounter and traverse peripheral blood capillaries. This parasite stage utilises an uncommon form of motility, termed gliding motility. It relies on a specialised, unconventional actomyosin motor system, situated below the plasma membrane, where the myosin powerstroke results in the rearward translocation of actin filaments (F-actin) and associated adhesins (Heintzelman, 2015). *Plasmodium* requires two highly sequence divergent actin isotypes for its cellular functions, with actin-1 being expressed throughout the life cycle and directly involved in gliding motility. Biochemically, actin-1 monomers assemble into F-actin at rates similar to vertebrate actin isotypes. However, *Plasmodium* F-actin appears to be dynamically unstable in vitro with very high disassembly and fragmentation rates (Schmitz et al, 2005; Vahokoski et al, 2014; Lu et al, 2019; Kumpula et al, 2017). Within the parasite, actin filaments have historically been difficult to visualise and the failure of traditional actin labelling tools on this divergent actin, has limited our understanding of dynamics within the cellular context. Recent work in *Plasmodium* and its related apicomplexan *Toxoplasma*, using the filament recognising actin chromobody, revealed localisations of actin filament pools primarily at the front (apical), rear (basal) and nuclear region of motile cells (Yee et al, 2022; Del Rosario et al, 2019; Tosetti et al, 2019). However, resolving these enigmatic actin filaments has proven difficult and an in vitro understanding of the arrangement, lengths, journey and fate of filaments in highly motile *Plasmodium* sporozoites remains unclear.

## Results and discussion

We used Focused Ion Beam milling (FIB-milling) and electron cryo-tomography (cryo-ET) to image actin filaments and other subcellular structures in *Plasmodium falciparum* sporozoites (Fig. EV1). Subvolume averaging (SVA) was used to determine the structure and a complete 3D model of F-actin within sporozoites (Figs. 1 and EV2). In total, volumes corresponding to ~85 individual cells from 29 tomograms were processed. F-actin was present in all major subcellular compartments (confirmed by SVA of individual filaments/compartments, Fig. EV3): the pellicular space (the intermembrane space between the plasma membrane and the inner membrane complex and the primary site for gliding machinery), the cytosol, and most remarkably in the nucleus. Surprisingly, and unlike some previous in vitro reports (Schmitz et al, 2005; Vahokoski et al, 2014; Kumpula et al, 2017), we consistently observed actin filaments longer than 100 nm, some up to 850 nm long (Figs. 1g and EV2b). The mean length of $200 \pm 140$ nm (standard deviation) is likely an underestimate due to

[1]Leibniz-Institut für Virologie (LIV), Hamburg 20251, Germany. [2]Centre for Structural Systems Biology, Hamburg 22607, Germany. [3]Department of Biochemistry, University of Oxford, South Parks Road, Oxford OX1 3QU, UK. [4]Department of Chemistry, Universität Hamburg, Hamburg 20148, Germany. [5]Biochemistry and Molecular Biology, Interdisciplinary Research Centre and Molecular Infection Biology, Biomedical Research Centre Seltersberg, Justus Liebig University Giessen, Giessen 35392, Germany. [6]Institute of Veterinary Physiology and Biochemistry, Justus Liebig University Giessen, Giessen 35392, Germany. [7]Institute for Structural and Molecular Biology, Division of Biosciences, University College London, London WC1E 6BT, UK. ✉E-mail: josie.ferreira@ucl.ac.uk

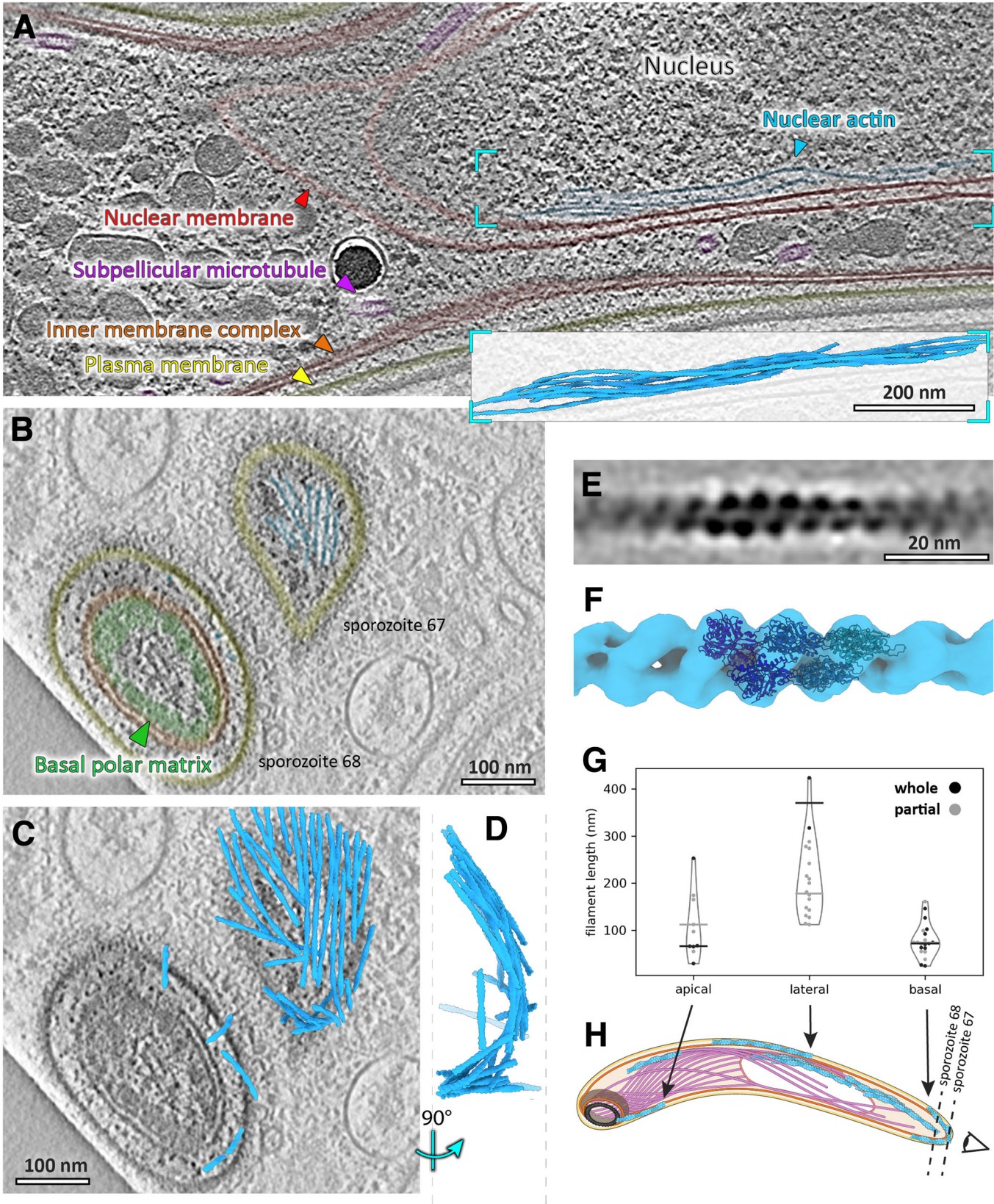

© The Author(s)

**Figure 1.   Discrete F-actin populations are found in *P. falciparum* sporozoites.**

(**A**) Slice through a tomogram showing a bundle of actin filaments in the nucleus. Inset shows a 3D representation of the actin bundle derived from subvolume averaging. (**B**) Slice through the basal ends of two sporozoites (see (**H**) for positioning of the slicing planes relative to the cells). (**C**) Same as (**B**) but overlaid with a 3D representation of actin from the whole tomogram. (**D**) Volume in (**C**) seen from the side. (**E**) Slice through the average volume of sporozoite F-actin. (**F**) Isosurface representation of the volume in (**E**) fitted with the molecular model of PDB 6TU4 (Vahokoski et al, 2022). (**G**) Size distribution of pellicular F-actin at different subcellular regions. Filaments that were fully contained within tomograms are shown in black, whereas those that were cut off by FIB-milling are shown in grey. Bars represent medians. N = 48 filaments. (**H**) Cartoon representation of a sporozoite cell with colours corresponding to structures labelled in (**A**) and dotted lines showing lamella orientations for cells 67 and 68 (-85 total) in (**B**) and (**C**). Source data are available online for this figure.

some filaments being truncated by FIB-milling. The global F-actin concentration was measured to be $40 \pm 7 \,\mu M$ (see Methods).

Analysis of the length distribution of pellicular F-actin shows two distinct length regulatory steps. (i) As filaments are transported down the cell towards the basal pole, they increase in length (Fig. 1G) and (ii) once at the basal pole, actin filaments are disassembled into shorter filaments (Fig. 1B–D,G). This observation of short filaments at the basal pole supports a recently proposed model of severing mediated by actin binding protein coronin, likely through recruitment of actin depoly-merizing factors (ADFs) (Douglas et al, 2018), as observed for other eukaryotes. In this mechanism, initial binding by coronin to the actin filament can either shield or allow access of ADFs, thereby enhancing severing and thus filament turnover (Gandhi et al, 2009; Galkin et al, 2011; Jansen et al, 2015; Mikati et al, 2015; Ge et al, 2014). The presence of coronin at the basal end of gliding *P. berghei* sporozoites (Bane et al, 2016), the observation that coronin overexpression rescued a gliding phenotype in an actin filament stabilized sporozoite (Yee et al, 2022; Douglas et al, 2018) and the observation of shorter filaments at the basal end in comparison with middle parts of the sporozoite (this study), further suggests that such a mechanism could be employed in sporozoites. The rate of filament disassembly is slower than the rate of filament accumulation at the basal pole, and in some cases filaments accumulate at the basal pole in a one-filament deep shell, suggesting that actin disassembly may be a rate-limiting step in motility (Fig. 1B,C). Previous results from actin filament stabilisation mutants show that filament stabilisation has an influence on turnover and thus has consequences for continuous motility (Douglas et al, 2018). As actin disassembly occurs within the restricted pellicular space, a local actin monomer gradient likely forms within this space. The path of glideosome actin filaments towards the basal pole, may therefore be up an actin monomer gradient. The age of the filament together with the action of formins and this gradient could account for the elongation we observe as filaments move down the cell, prior to their disassembly at the basal pole. Localised pools of actin-stabilising proteins may also contribute to the length increase we observe.

Our data suggests that the apical and basal poles are regulatory sites for pellicular actin. At the apical pole, we observed several filaments in close proximity to the preconoidal rings (Fig. 2). Recent observations in related apicomplexans *Cryptosporidium parvum* and *Toxoplasma gondii* suggest that pellicular actin is nucleated at the preconoidal rings, implying that this is a consistent apicomplexan feature (Fig. 2A, Video 1) (Martinez et al, 2023; Dos Santos Pacheco et al, 2022). However, in *C. parvum* and *T. gondii*, the channeling of filaments into the pellicular space is dependent on extrusion of the conoid—a structural feature that is missing in *Plasmodium* sporozoites (Ferreira et al, 2023).

F-actin nucleation at the apical pole creates a constant flux of actin from the cytosol to the pellicular space. How then is the cytosolic actin pool recycled? The IMC in sporozoites has very few discontinuities,

apart from the basal pole, which was dotted with ~25 nm diameter pores (Figs. 2B and EV4). These pores were observed almost exclusively at the basal end, leading us to hypothesise that their localization was related to the unusually high concentration of basal F-actin. Strikingly, we observed filamentous actin protruding through or in direct proximity to basal pores ($N = 2$), suggesting that filamentous actin can pass through (Figs. 2B and EV4C, Videos 2, 3). It is therefore possible that these pores could facilitate more efficient exchange of actin between the pellicular space and the rest of the cell. This would recycle actin and lower the concentration of actin at the rear of the sporozoite (up to 60 mM local F-actin concentration 100 nm from the basal pole end, see methods) and thus could allow for more efficient gliding. The lack of similarly sized pores at other parts of the cell indicates that their location at the basal pole is regulated, facilitating the directional flow of actin into the pellicular space only via the preconoidal rings, and into the cytoplasm only via the basal pores. This is likely a conserved apicomplexan feature as similar pores were observed in *C. parvum* (Martinez et al, 2023), while *T. gondii* has a large opening in the IMC at the basal pole—both of which could facilitate directional actin flow. It would be reasonable to speculate that the posterior/basal polar ring complex (De Niz et al, 2017) could be involved in the localisation of the pores. Notably, we have not observed a ring-like structure at the basal pole, but rather a thick amorphous layer, which we refer to as basal polar matrix (Fig. EV4C).

While examining the glideosome's local environment, we noticed a network of thin filaments reinforcing the outer IMC membrane (membrane surface pointing towards the glideosome and plasma membrane, Figs. 2C–F and 3B,C, Video 4). We refer to these filaments in *P. falciparum* as thin pellicular filaments (TPFs). TPFs were always oriented parallel to the long parasite axis, with approximately 20 nm (~5–40 nm) interfilament spacing,  and with rare cross-links. A SVA of TPFs highlighted that they are embedded in the outer IMC membrane (Fig. 3C). In two cellular locations, we observed glideosomal actin filaments bound by densities consistent with myosin heads, with tails leading to TPFs in the IMC (Fig. 2F, Video 5). The observed features prompted us to hypothesise that TPFs are structural elements that allow the force generated by the glideosome to be distributed along the entire IMC. TPFs bear no ultrastructural similarity to IMC surface filaments (IMCFs) observed in *C. parvum*, which form tracks to guide actin filaments down the cell (Martinez et al, 2023). Although it has been suggested that the gliding machinery is directly connected to subpellicular microtubules (SPMTs, Harding et al, 2019), with our current approach, we see no direct connection between TPFs, actin and SPMTs through the IMC membranes (Fig. 3). Although we observe homogenous ~4 nm globular proteins that span the IMC mem-branes, and which we speculated may link the glideosome/TPFs to the SPMTS, these densities are not statistically associated with TPFs, indicating no interaction (Fig. 3B,D).

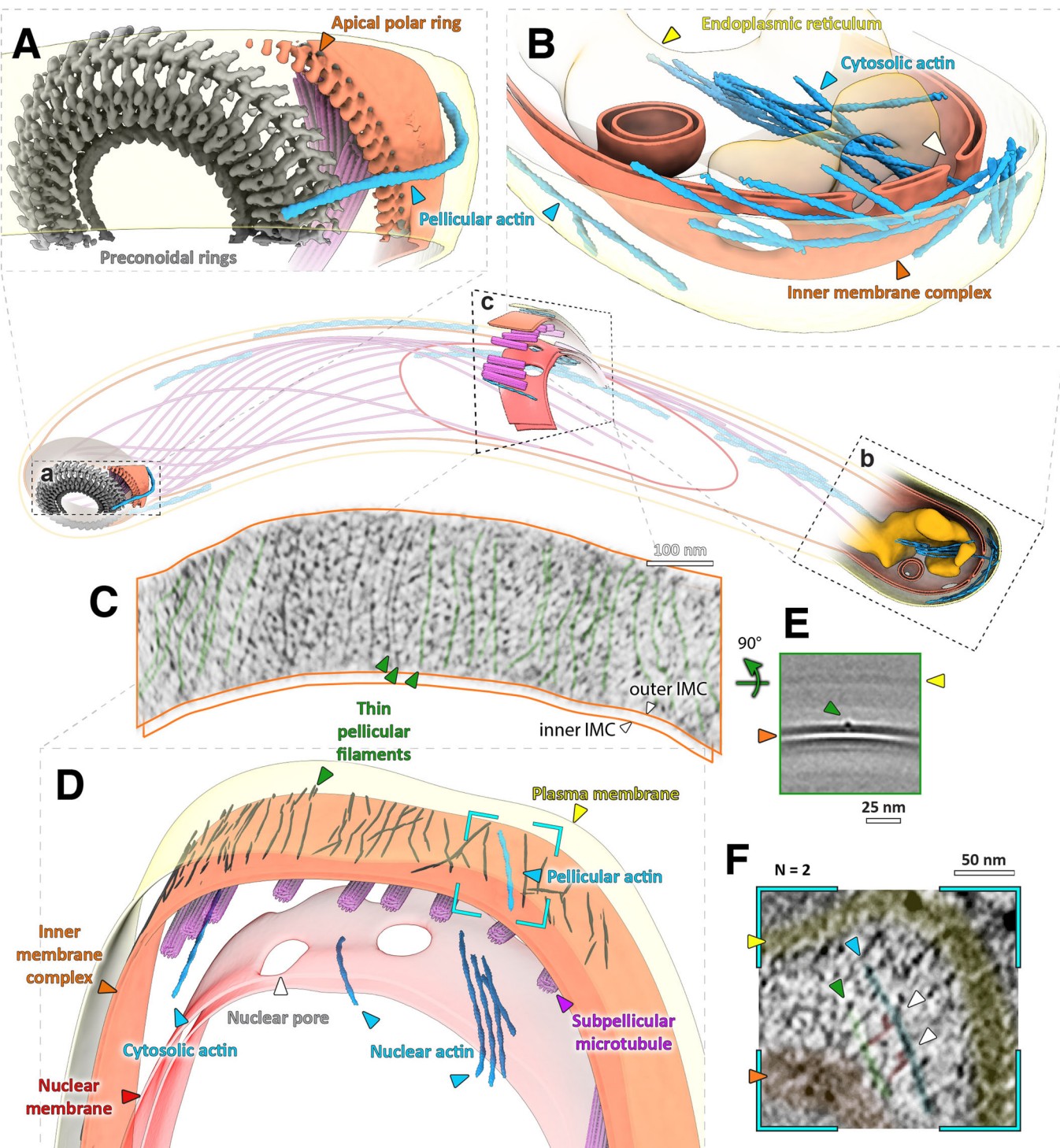

**Figure 2. Visualising discrete actin populations within the 3D cellular context of motile sporozoites.**

Central cartoon represents the approximate position of volumes (**A–C**) within a cell. (**A**) Apical end of showing a single actin filament being nucleated at the pre-conoidal rings (grey). (**B**) Basal end of a sporozoite, showing cytoplasmic actin bundles as well as a build-up of pellicular actin. White arrowhead indicates an actin filament going through a basal pore. Shown is also an invagination of the IMC (observed in three cells). See slice through this tomogram in Fig. EV4C. (**D**) A lateral section showing nuclear, cytosolic and pellicular F-actin. Dark lines on the outer surface of the IMC represent thin pellicular filaments (TPFs). (**C**) A slice along the surface of the outer leaflet of the outer IMC membrane. Some filaments have been highlighted in green. (**E**) A slice through the average volume of TPFs showing their position relative to the IMC membranes. Orthogonal sections through this volume also shown in Fig. 3C. (**F**) A slice through the tomogram (position shown in **D**) showing a pellicular actin filament (blue) connected to a TPF by two densities (red, white arrows) consistent with myosin dimensions (observed in two different locations). Source data are available online for this figure.

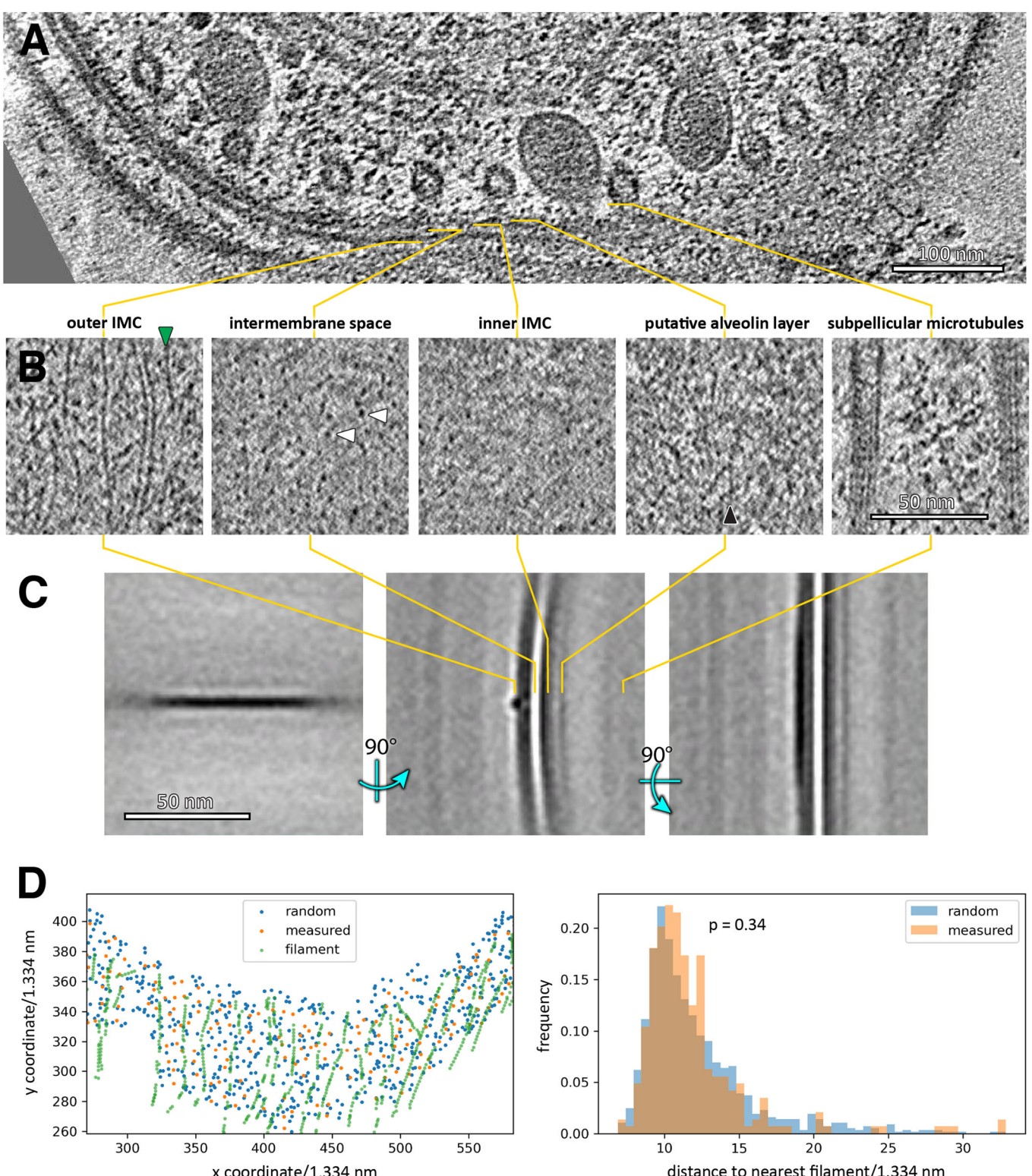

Apart from motility, F-actin is implicated in multiple cellular roles including intracellular transport, transcriptional regulation and cell structural support. We therefore analysed the populations of F-actin observed in compartments other than the pellicular space. There were numerous examples of F-actin in the cytoplasm of sporozoites

(Fig. 2B,D). These were found primarily as individual filaments, with some bundles (10 separate filaments and 4 bundles consisting of 2–8 filaments). F-actin was not associated in any obvious pattern with any subcellular element, but was typically oriented along the parasite's apico-basal axis. What was most surprising was the large amounts of F-actin

**Figure 3. Analysis of the glideosome's local environment; the pellicular space.**

(A) A slice through a tomogram (also in Fig. 2C) highlighting the pellicle ultrastructure. (B) Slices through tomogram shown in (A), oriented parallel to the inner membrane complex (IMC) at indicated relative positions. We observed a large number of globular particles spanning the space between the two IMC membranes, with a homogeneous size distribution (~4 nm diameter). (C) Orthogonal sections through an average volume of ~3000 thin pellicular filament particles. Middle section also shown in Fig. 2E. (D) We measured the distance distribution of the intermembrane particles to the nearest TPF to determine whether these may be directly interacting. However, a randomly distributed set of coordinates (on a surface defined by the measured intermembrane particle coordinates, left) has the same distribution of distances (right), indicating that no interaction takes place. Population distributions were compared using a Mann–Whitney U Test. Source data are available online for this figure.

we observed within the nucleus (Figs. 1A, 2D, Videos 4, 6, 7). In fact, the nucleus contains the majority of observed F-actin in sporozoites: approximately 60% (controlled for subcellular areas imaged in tomograms). It was found predominantly in bundles of ~3–8 filaments with at least one filament less than 10 nm from the nuclear membrane (global median distance of all filaments to the inner nuclear membrane was 21 ± 11 nm, Fig. 1A). Whether this is due to specific binding to a component of the membrane or due to marginalisation is not clear. While there have been many reports of the presence of nuclear actin in different organisms, this is to our knowledge, the first direct evidence of nuclear F-actin in the absence of staining or stabilising agents in any organism. Indeed, actin signals have indicated F-actin accumulation near to the nucleus (in ~20% of sporozoites in the case of the actin chromobody) during motility and invasion of apicomplexan parasites, suggesting that a nuclear actin cage facilitates efficient invasion and/or protects the nucleus from damage when the parasite undergoes constriction (Yee et al, 2022; Del Rosario et al, 2019; Angrisano et al, 2012). However, here we have observed extensive bundles within the nucleus itself (Figs. 1A and 2D). *Plasmodium* actin has previously been implicated in spatial repositioning of genes and histone methylation states in monoallelic expression of blood stages (Zhang et al, 2011; Volz et al, 2012). Whether these actin bundles observed in sporozoites serve as an intra-nuclear protective or mechanosensing structure, or fulfil additional molecular roles in gene expression in the nucleus requires further investigation.

Actin is key for the transmission of the malaria parasite. By dissecting the path of F-actin within the glideosome we provide new mechanistic understanding of the dynamics of filament transport down the cell, severing at the basal pole and transfer, and recycling via IMC pores. The optimal structural preservation and high resolution in situ data presented here will provide a framework for integrating future findings from other, more reductionist methods, in order to gain a complete understanding of the roles of the malaria parasite's actin cytoskeleton in motility, in the cytoplasm, and in the nucleus.

## Methods

### Reagents and tools table

| Reagent/Resource | Reference or Source | Identifier or Catalog Number |
| --- | --- | --- |
| **Experimental models** | | |
| Plasmodium falciparum sporozoites | TropIQ https://tropiq.nl/ | NF54-ΔPf47-5′ csp-GFP-Luc |
| **Chemicals, Enzymes and other reagents** | | |
| L-15 Leibovitz medium | Lonza | Cat# 12-700F |
| RPMI – phenol red | Gibco | Cat# 11835030 |

| Reagent/Resource | Reference or Source | Identifier or Catalog Number |
| --- | --- | --- |
| UltrAufoil R1.2/1.3 300 mesh EM grids | Quantifoil https://www.quantifoil.com/products/ultraufoil | N1-A14nAu30-01 |
| **Software** | | |
| MAPS | Thermofisher scientific | |
| SerialEM | https://bio3d.colorado.edu/SerialEM/ Mastronarde (1997) | |
| IMOD | https://bio3d.colorado.edu/imod/ Kremer et al (1996) | |
| PEET | https://bio3d.colorado.edu/PEET/ Heumann et al (2011) | |
| TEMPy | https://tempy.topf-group.com/ Cragnolini et al (2021) | |
| Scipy | https://scipy.org/ Virtanen et al, 2020 | |
| Scikit-learn | https://scikit-learn.org/stable/ Pedregosa et al, 2011 | |
| Matplotlib | https://matplotlib.org/ Hunter, 2007 | |
| Numpy | https://numpy.org/ Harris et al, 2020 | |
| Python3 | https://www.python.org/download/releases/3.0/ | |
| Bsoft | https://cbiit.github.io/Bsoft/ Heymann, 2001 | |
| UCSF ChimeraX | https://www.cgl.ucsf.edu/chimerax/ Pettersen et al, 2021 | |
| Open3d | https://www.open3d.org/ preprint: Zhou et al, 2018 | |
| **Other** | | |
| Aquilos 2 | Thermofisher scientific | |

### Obtaining sporozoites

*P. falciparum* sporozoites (strain: NF54-ΔPf47-5′csp-GFP-Luc: expressing a GFP-Luciferase fusion protein under the control of the csp promoter, genomic integration, no selection marker) were prepared at TropIQ (Nijmegen, Netherlands). Gametocytes were fed to 2 day old female *Anopheles stephensi* mosquitoes. Mosquito infection was confirmed 7 days post feeding by midgut dissection. At 7 days post infection, mosquitoes received an extra non-infectious blood meal to boost sporozoite production. Two weeks

post infection, sporozoites were isolated using salivary gland dissection and released using mechanical crushing into Leibovitz medium supplemented with 10% heat inactivated human serum, prior to shipping at room temperature.

## Cryo-grid preparation

*P. falciparum* sporozoites were checked under the fluorescent microscope and then diluted 1:4 into RPMI medium (without phenol red). 3 µl of parasites were applied onto a freshly plasma-cleaned UltrAufoil R1.2/1.3 300 mesh EM grid (Quantifoil) in a humidity controlled facility. Excess liquid was manually back-blotted and grids were plunged into a reservoir of ethane/propane using a manual plunger. Grids were stored under liquid nitrogen until imaging. Two replicates were performed from independent shipments (each made up of sporozoites dissected from ~25 mosquitoes).

## Cryo FIB-milling

Grids were clipped into autogrids modified for FIB preparation (Schaffer et al, 2015) and loaded into either an Aquilos or an upgraded Aquilos2 cryo-FIB/SEM dual-beam microscope (Thermofisher Scientific). Overview tile sets were recorded using MAPS software (Thermofisher Scientific) before being sputter coated with a thin layer of platinum. Good sites with parasites were identified for lamella preparation before the coincident point between the electron beam and the ion beam was determined for each point by stage tilt. Prior to milling, an organometallic platinum layer was deposited onto the grids using a GIS (gas-injection-system). Lamellae were milled manually until under 300 nm in a stepwise series of decreasing currents. Milling was performed at the lowest possible angles to increase lamella length in thin cells. Finally, polishing of all lamella was done at the end of the session as quickly as possible but always within 1.5 h to limit ice contamination from water deposition on the surface of the lamellae. Before removing the samples, the grids were sputter coated with a final thin layer of platinum. Grids were stored in liquid nitrogen for a maximum of 2 weeks before imaging in the TEM.

## Tilt-series collection

Cryo-EM FIB-milled grids were rotated by 90° and loaded into a Titan Krios microscope (Thermofisher) equipped with a K3 direct electron detector and (Bio-) Quantum energy filter (Gatan). Tomographic data was collected with SerialEM with the energy-selecting slit set to 20 eV. Datasets were collected using the dose-symmetric acquisition scheme at a ±65° tilt range with 3° tilt increments. For all datasets, 5–10 frames were collected and aligned on the fly using SerialEM and the total fluence was kept to less than 120 e$^-$Å$^2$. Defoci between 3 and 8 µm underfocus were used to record the tilt series'.

## Tomogram reconstruction

Frames were aligned on the fly in SerialEM (Mastronarde, 1997); CTF estimation, phase flipping and dose-weighting was performed in IMOD (Kremer et al, 1996). Tilt-series' were aligned in IMOD either using patch-tracking or by using nanoparticles (likely gold or platinum) on lamella surfaces as fiducial markers. Tomograms were binned 4x and filtered in IMOD or by using Bsoft (Heymann, 2001).

## Subvolume averaging

Subvolume averaging was performed using PEET (Heumann et al, 2011) as described previously (Ferreira et al, 2023). Model processing was done using TEMPy (Cragnolini et al, 2021), Scipy (Virtanen et al, 2020), Scikit-learn (Pedregosa et al, 2011), Matplotlib (Hunter, 2007) and Numpy (Harris et al, 2020) in Python 3. Initial models were generated manually by picking line segments using pairs of IMOD model points and then interpolating particles at 1 voxel (1.3 nm) increments. The initial Y axes were aligned with the line segments and Y axis rotation angles were randomised. The initial reference was generated by averaging particles with the starting orientations, thus generating a featureless cylinder. A small subset of particles (~700) were refined to create a reference with F-actin features which was then used for alignment of ~70k initial positions. Duplicate and low-scoring particles were removed. In order to improve model completeness and allow separation of particles into two independent halves, the subvolume positions were then fitted to a spline-smoothed helical model allowing for small variation in helical pitch (Fig. EV2). Subvolume positions were then generated based on the best fitting model parameters. These were split into two halves and aligned independently. Overlapping particles between the two half-maps were removed before generating final half-maps. Fourier Shell Correlation was measured using Bsoft, suggesting 27 Å resolution at the 0.143 cutoff. Particles from the two half-datasets (11487 total) were then combined and aligned together. The final volume was sharpened using Bsoft with an arbitrarily chosen B-factor of −3000 for fitting and visualisation.

## Segmentation and visualisation

Membrane segmentation was performed in IMOD, using drawing tools followed by linear interpolation. These were then resampled using open3d to achieve an isotropic coordinate distribution, which were then used to generate a volume using IMOD imodmop. F-actin, microtubules, apical polar ring and preconoidal rings were backplotted: average volumes were placed into 3D volumes using coordinates determined by SVA. Actin and microtubule models were smoothed for backplotting. Surface visualisation was performed using UCSF ChimeraX (Pettersen et al, 2021) or open3d (preprint: Zhou et al, 2018). Volume sections were visualised using IMOD 3dmod. Plots were generated using Matplotlib.

## Length measurements

Filament lengths for comparison of nuclear, cytosolic and pellicular filament lengths were derived from helical models based on subvolume averaging positions (see above). Filament lengths for comparison of apical, lateral and basal pellicular filament lengths were measured manually using 3dmod.

## Thin pellicular filament interaction analysis

To evaluate whether thin pellicular filaments (TPFs) are physically interacting with the large globular particles spanning the IMC intermembrane space, we compared their distance distribution to a random population: First, a point cloud representing the inner IMC membrane was generated using Open3d following manual

segmentation with ~10 points per pixel. The centroid of each intermembrane particle ($N = 271$) was determined using subvolume averaging, following manual picking. These 3D coordinates (here referred to as $C_{data}$) were projected onto the membrane plane (via nearest neighbour) and their distance ($D_{data}$) to the nearest TPF particle ($N = 1333$) was measured using Scipy.spatial.KDTree. The median $D_{data}$ was 8.3 nm. A random set of membrane coordinates ($C_r$) was selected to obtain an equal sample size and a similar distribution: $C_{random}$ were no further than 11 nm from a data coordinate (mean $C_{data}$ spacing plus two times the standard deviation). The $C_{random}$ distance ($D_{random}$) to the nearest TPF was measured (8.3 nm median). $D_{data}$ and $D_{random}$ were compared using Mann–Whitney U Test (scipy.stats.mannwhitneyu) as the normality of both distribution differed significantly from normal when measured by scipy.stats.normaltest ($p$ values $4.8 \times 10^{-149}$ and $4.9 \times 10^{-46}$, respectively).

## F-actin concentration

The number of actin subunits in observed F-actin was estimated from subvolume averaging (15,058) and manual length measurements (17,165, assuming 38 nm per 13 subunits). The subvolume averaging-derived value is likely an underestimate due to cross-correlation-based particle cleaning; it is the number of particles after the first alignment step of the two independent datasets. The two estimates were used to calculate the experimental error, expressed as standard deviation. The total observed volume of 29 tomograms with an average thickness of 244 nm was $4.2 \times 10{-17}$ m$^3$, of which cells made up approximately 7/12. $9.7 \times 10^{-19}$ mol in $2.4 \times 10^{-14}$ L corresponds to $4.0 \times 10^{-5}$ mol L$^{-1}$. For local actin concentration, the volume of the pellicular space was measured using manually segmented models at the basal end of a sporozoite oriented roughly perpendicularly to the FIB-sectioning axis (sporozoite 1, Fig. 1C,H). $8.73 \times 10^{-20}$ moles of actin (based on subvolume averaging data) in $1.53 \times 10^{-18}$ L of pellicular space corresponds to a local concentration of $5.7 \times 10^{-2}$ mol L$^{-1} \sim 60$ mM.

## Data availability

Subvolume average has been deposited on the EMDB (EMD-19898): https://www.ebi.ac.uk/emdb/EMD-19898.

The source data of this paper are collected in the following database record: biostudies:S-SCDT-10_1038-S44319-025-00415-7.

## Peer review information

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

## Acknowledgements

We thank Lindsay Baker for helpful discussions and Carolyn Moores for her continued support and critical reading of the manuscript. Thank you to the CSSB EM facility team for their support; this research was funded in part by DFG INST 152/ 772-1, 774-1, 775-1, 777-1 FUGG (CSSB cryoEM facility). We gratefully acknowledge funding by HFSP long-term postdoctoral fellowship LT000024/2020-L (JLF), Infrastructures for the control of vector-borne diseases (Infravec2) funded by the EU's Horizon 2020 programme (grant agreement No 731060) (JLF), Wellcome Career Development award 227774/ Z/23/Z (JLF), LOEWE Centre DRUID ("Novel Drugs Targets against Poverty-related and Neglected Tropical Infectious Diseases") within the Hessian Excellence Program (RGD). Wellcome Trust 209250/Z/17/ Z and 107806/Z/ 15/Z (KG). For the purpose of open access, the author has applied a Creative Commons Attribution (CC BY) licence to any Author Accepted Manuscript version arising.

## Author contributions

**Vojtech Pražák**: Conceptualization; Data curation; Software; Formal analysis; Validation; Investigation; Visualization; Methodology; Writing—original draft; Writing—review and editing. **Daven Vasishtan**: Software; Validation. **Kay Grünewald**: Resources; Supervision; Funding acquisition; Writing—review and editing. **Ross G Douglas**: Conceptualization; Data curation; Writing—original draft; Writing—review and editing. **Josie L Ferreira**: Conceptualization; Funding acquisition; Validation; Investigation; Methodology; Writing—original draft; Project administration; Writing—review and editing.

Source data underlying figure panels in this paper may have individual authorship assigned. Where available, figure panel/source data authorship is listed in the following database record: biostudies:S-SCDT-10_1038-S44319-025-00415-7.

## Disclosure and competing interests statement

The authors declare no competing interests.

# Expanded View Figures

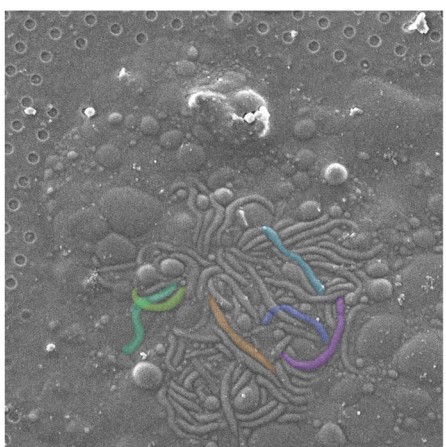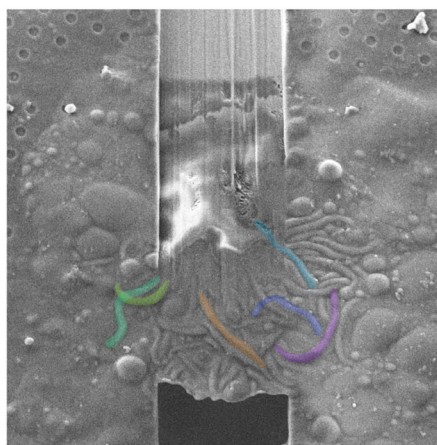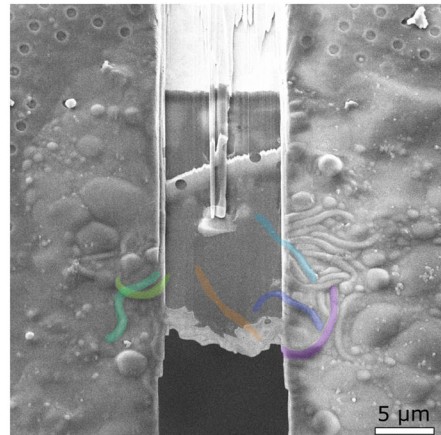

**Figure EV1.  FIB-milling sporozoites.**

Left: A pile of sporozoites in the SEM. A few individual sporozoites are coloured consistently in all three images. Middle: SEM image during the FIB-milling process. Right: SEM image of the final polished lamella. Source data are available online for this figure.

**A**

1) Reference generation: ~ 700 particles

 i) Initial reference

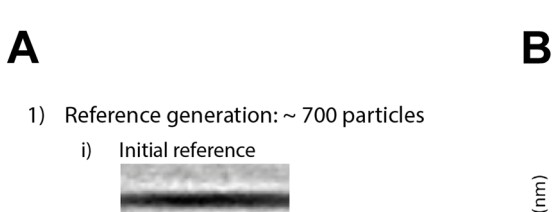

 ii) After 5 round of alignment

2) Template matching

3) Helical fitting

4) "Gold standard" alignment

 i) Initial reference: 7679 particles (half-map)

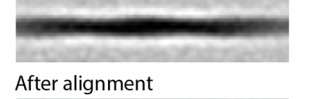

 ii) After alignment

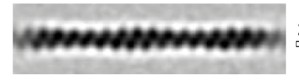

20 nm

 iii) Shorter mask

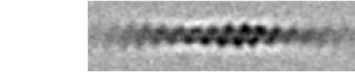

**B**

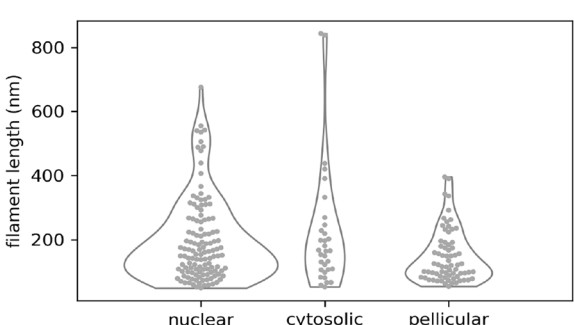

**C**

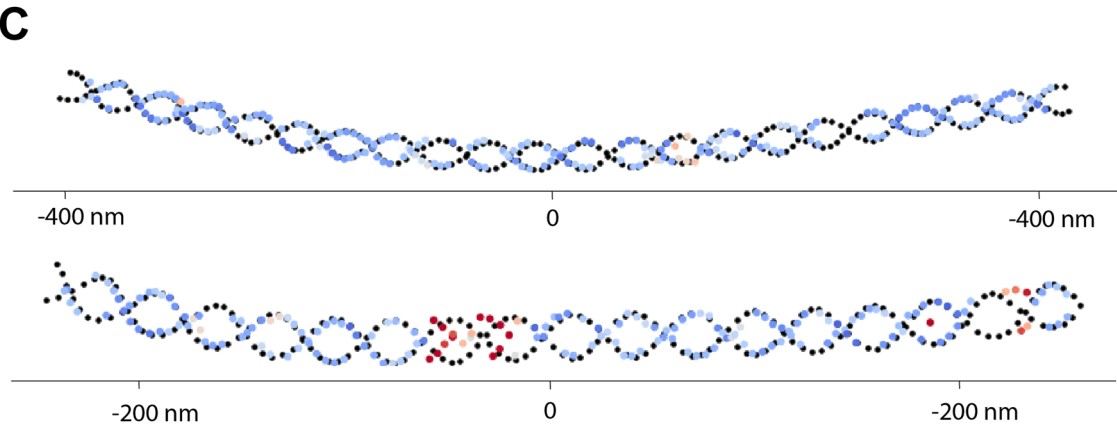

Figure EV2. Subvolume averaging of actin.

(A) A rough outline of the workflow used for subvolume averaging of actin, with some reference volumes shown. (B) Size distribution of F-actin in indicated subcellular locations derived from subvolume averaging coordinates. Note that especially longer filaments were frequently truncated during FIB-milling causing the distribution to be skewed towards shorter sizes. Two-sided t-test $p$-values are 0.4 and $1 \times 10^{-4}$ comparing nuclear and cytosolic, and nuclear and pellicular, respectively. Caution should be used interpreting the significance of the difference due to some filaments being truncated. (C) Diagnostic plots of helical fitting of two longer filaments. Black dots show best fit positions of actin subunits. Overlaid are positions measured by subvolume averaging coloured from blue to red based on distance to the nearest model point. Source data are available online for this figure.

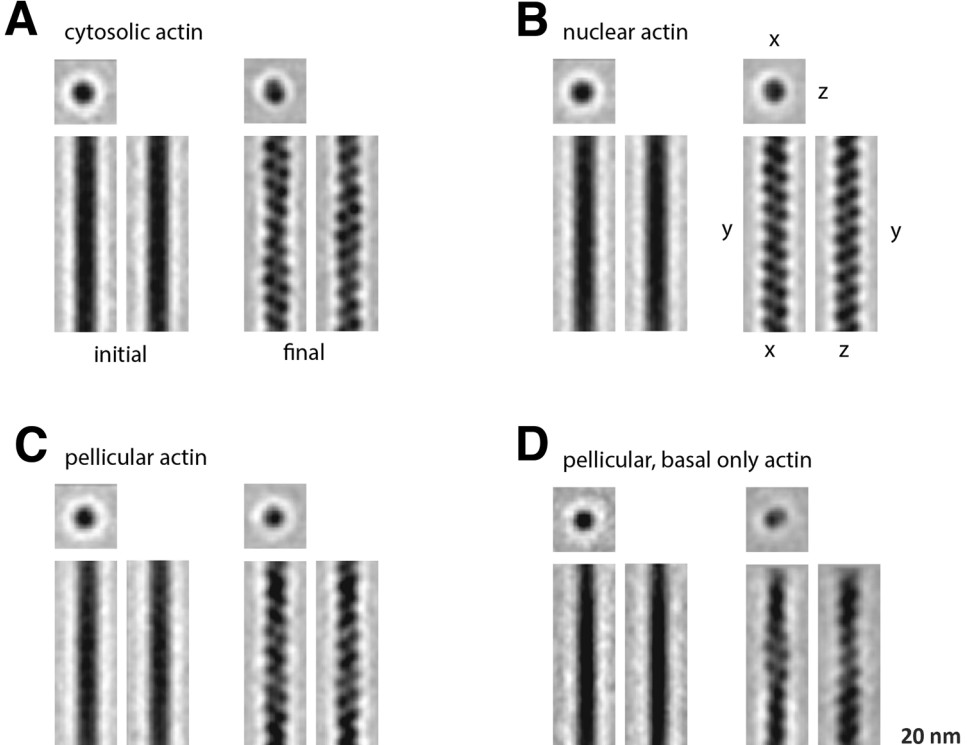

**Figure EV3. Subvolume averages of actin from different subcellular compartments.**

Actin filaments used for subvolume averaging (Figs. 1E and EV2) were manually split into classes based on their subcellular localization: (**A**) cytosol, (**B**) nucleus, (**C**) pellicular space, (**D**) pellicular space at the basal end. The particle Y axis (long filament axis) orientations were randomised and independent references were generated for each class (left hand side). Alignment was performed without refining orientations. Source data are available online for this figure.

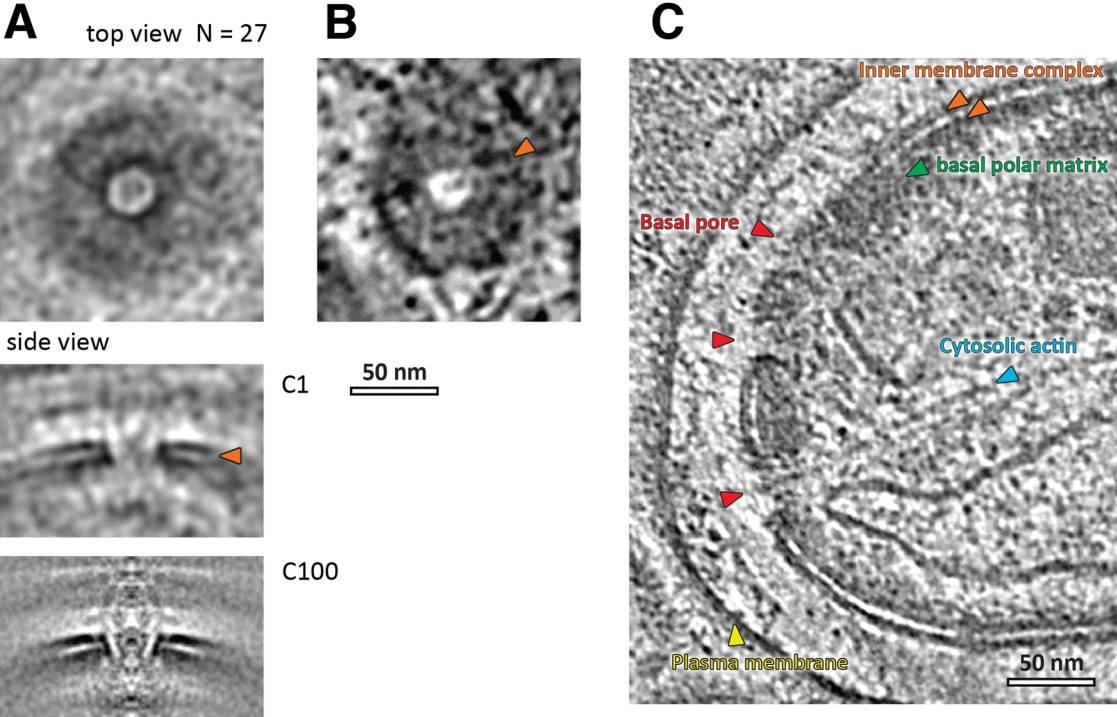

**Figure EV4. Basal pores form 25 nm diameter connections between the cytoplasm and pellicular space.**

(A) Subvolume average of 27 particles. No symmetry is evident at this resolution. (B) A tangential slice through a single pore. (C) Slice through a basal end of a sporozoite (also in Fig. 2B) showing three basal pores. Source data are available online for this figure.

