## [Peer Review File · EMBO Reports]

Molecular architecture of glideosome and nuclear F-actin in *Plasmodium falciparum*

Vojtech Prazak, Daven Vasishtan, Kay Grunewald, Ross Douglas, and Josie Ferreira

Corresponding author(s): Josie Ferreira (josie.ferreira@ucl.ac.uk)

Review Timeline:

Submission Date:	31st Oct 24
Editorial Decision:	20th Dec 24
Revision Received:	30th Jan 25
Accepted:	21st Feb 25

Editor: Deniz Senyilmaz Tiebe

Transaction Report: This manuscript was transferred to EMBO reports following peer review at Review Commons.

**Review
COMMONS**

Review #1

1. Evidence, reproducibility and clarity:

Evidence, reproducibility and clarity (Required)

Pražák et al.: Molecular architecture of glideosome and nuclear F- actin in Plasmodium falciparum

The study by Pražák et al. uses cryo-electron microscopy to explore unresolved questions about the cellular and molecular architecture of the actin cytoskeleton in motile malaria parasite Plasmodium falciparum sporozoites (liver-infective forms). The key focus of the study is to determine the localization and organisation of actin filaments (F-actin). Employing a powerful combination of focused ion beam milling and electron cryo-tomography, the researchers visualize intact F-actin in sporozoites for the first time, detailing their organisation and potential journey, using this to define a model for their formation, progress and ultimate recycling within sporozoites (in relation to gliding motility/the sub-pellicular space) as well as observing actin in the nuclear periphery. The study reveals actin filaments that are longer than previously observed in vitro and how they concentrate organise within the cell in their "native" state.

****Key Insights that are notable in the study:****

- Visualization of F-actin: The authors present direct visualization of actin filaments in different phases of growth within Plasmodium falciparum cells, showing filaments much longer than previously seen in vitro (near micron in length rather than 100s of nm).
- Nuclear Actin Bundles: The authors observe extensive actin bundles within the nucleus, suggesting potential undefined roles.
- Pellicular Intermediate Filaments (PIFs): The authors identify a novel class of undefined filament - calling them PIFs - with no direct connection to subpellicular microtubules being observed.
- Actin Filament Dynamics: The authors present their data in the context of a well-accepted model for actin filament dynamics with assembly at the apical end, growth as they are transported down the cell, and disassembly at the basal end.
- Actin Exchange and Recycling: The authors observe large pores at the basal end of the IMC that they postulate these facilitate actin exchange between the pellicular space and cytosol, helping to maintain directional actin flow for efficient gliding motility.

****Major comment:****

- The study is beautifully executed and will clearly generate a lot of interest in the gliding/cytoskeletal field. I applaud the authors for that. My chief concern is, however, the numbers of cells involved AND whether the authors can be sure that their observations provide compelling evidence for the conclusions they make OR are the basis for limited hypothesis only. Most of my comments below lead directly from this concern about number of cells and a tendency (which I understand) to build hypotheses on the data observed. I feel at times the authors veer to close to speculation without drawing enough from the literature that has come before.

Questions:

1. "Subvolume averaging (SVA) was used to determine the structure and a complete 3D model of F-actin within sporozoites"

- To my reading, though I am uncertain, the entire paper is based on only 2 sporozoites? At least this is not qualified anywhere in the paper I can see easily (apologies if I have missed it). This needs clarification in the methodology and results sections.

- Assuming that few cells are the basis for observations, the conclusions made I would argue should be presented as speculative (e.g. based low numbers, n, of cells and individual observations). How many cells were used?

- Also - sporozoites in the salivary gland (the origin of these cells) are relatively immotile until they are released. Do the authors have data on whether the sporozoites were active or not when harvested? This will have implications for the state of F-actin observed and how it relates to gliding.

2. "a recently proposed model of severing mediated by coronin, an actin binding protein located at the basal end of gliding *P. berghei* sporozoites¹³".

- Coronin is not generally thought of as a disassembly factor (though it may be a minor player in this space). The ref cited (13, Bane et al.) demonstrates that following activation, coronin binds to actin filaments and then tracks with the protein to the rear end of the sporozoite. This fits its F-actin binding properties & membrane binding properties. I.e. it might just track engaged filaments. But my re-reading of the Bane paper does not suggest a model for coronin-dependent filament disassembly as far as I can see.

- What about membrane dynamics in general (plasma membrane retrograde flow <https://pubmed.ncbi.nlm.nih.gov/26792112/>)? This is largely absent from discussion in the paper.

- Much more likely is ADF/cofilin playing a role in disassembly - though its generally cytoplasmic distribution (noted in *Toxoplasma* and *Plasmodium*) makes it hard to resolve exactly where it asserts its function.

3. "at the basal pole, and in some cases form a one-filament deep shell, suggesting that actin disassembly may be rate-limiting".

- It's fine to speculate that shorter filaments at the basal pole is consistent with a model where this is the site of disassembly (no-brainer) but I don't understand the comment about rate limiting? What do the authors mean by this? The speed of disassembly limits the speed of the motile zoite? If so, what is the evidence that supports this beyond a simple accumulation of filaments here. That wasn't clear to me.

4. "The path of glideosome actin filaments towards the basal pole, may therefore be up an actin monomer gradient. The age of the filament together with this gradient could account for the elongation we observe as filaments move down the cell, prior to their disassembly at the basal pole. Localised pools of actin-stabilising proteins may also contribute to the length increase we observe."

- The missing piece of the puzzle here is nucleation. Where does it occur and does it continue down the length of the sporozoite. Formins (the primary nucleator in *Plasmodium*) concentrate at the sporozoite apex, but they may also track down the cell/IMC space. Since Formin's bind the barbed end (growth end) they can continue to grow filaments as they might migrate (if they stay with the filament). I feel the above sentence is predominantly speculation - the evidence from the imaging does not point to any model but supports data from other studies.

5. "We observed filamentous actin protruding through some basal pores, suggesting that

filamentous actin can pass through into the cytoplasm... possible that these pores could facilitate more efficient exchange of G- and F-actin between the pellicular space and the rest of the cell."

- How many events is this observation based on? Are the authors sure that in the compendium of images by EM from the literature (from the likes of Bannister, Aikawa, Cyrklaff, Ferguson etc.) a) pores haven't already been described at the basal region of the IMC in plasmodium and b) that the odd structure consistent with an F-actin filament wasn't documented? This would make their argument far more convincing. If it's $n=1/2$ then it's quite speculative.

6. "lower the concentration of actin at the rear of the sporozoite (up to 60 mM local F-actin concentration 100 nm from the basal pole end) and thus could allow for more efficient gliding"

- Where does this number 60 come from?

7. "In two cellular locations we observed glideosomal actin filaments bound by densities consistent with myosin heads, with tails leading to PIFs in the IMC".

- I would love this to be the case (many in the field have long sought to visualise the motor).

However, if this is simply based on $n=2$ densities (in 1 cell?) it feels very speculative.

8. "While there have been many reports of the presence of nuclear actin in different organisms, this is to our knowledge, the first direct evidence of nuclear F-actin in the absence of staining or stabilising agents in any organism."

- The caveat at the end of this sentence is a bit unreasonable to my mind. Back in 2008 Cryo-SEM showed F-actin in the nuclear periphery of *Xenopus* oocytes

(<https://pubmed.ncbi.nlm.nih.gov/19017237/>). It is true this requires some level of shadowing etc.

but is that labelling? Does it matter? There is a whole field of research exploring nuclear actin across eukaryotes which feels a bit overlooked here.

9. "Whether this simply serves as an intra-nuclear protective structure or fulfils additional molecular roles, such as mechanosensing, in the nucleus requires further investigation."

- Actin has been implicated in numerous functions in Apicomplexa and the authors would do well to cite much more of the literature, otherwise there is a sense that this is entirely uncharted territory. The work of the Scherf lab (e.g. <https://pubmed.ncbi.nlm.nih.gov/22100161/>), Cowman lab (e.g. <https://pubmed.ncbi.nlm.nih.gov/22264509/>) and my own lab (e.g.

<https://pubmed.ncbi.nlm.nih.gov/22389687/>) all discuss potential roles for nuclear actin that are worth considering in the context of having imaged it with new resolution.

****Minor Comment:****

In the supplementary movies, could the authors not include the (now standard) cryo-EM fly through where the sections are transitioned through with addition of rendering to build the 3D models shown in the body of the paper. This would give a much clearer interpretation of where F-actin is.

2. Significance:

Significance (Required)

- There is the nugget of something very important in this paper. My chief concern is the limitations that would come from the numbers of cells imaged (ensuring the data is properly quantified), and whether this is substantial enough to then interpret and build robust hypotheses from.

- With robust data (i.e. many cells, perhaps in different states of motility/non-motility) this paper would then become the foundation for others to go and test (at molecular and cellular levels)

whether the hypotheses raised are correct and ultimately resolve the key question - what is F-actin doing in the sporozoite and how it is regulated?

- The cell biology/parasitology community would be very interested in such a paper!

- I have had a long-standing interest in the cell biology of apicomplexan parasites (plasmodium specifically) and worked for many years on the actomyosin gliding motor.

3. How much time do you estimate the authors will need to complete the suggested revisions:

Estimated time to Complete Revisions (Required)

(Decision Recommendation)

Between 1 and 3 months

4. Review Commons values the work of reviewers and encourages them to get credit for their work. Select 'Yes' below to register your reviewing activity at Web of Science Reviewer Recognition Service (formerly Publons); note that the content of your review will not be visible on Web of Science.

Yes

Review #2

1. Evidence, reproducibility and clarity:

Evidence, reproducibility and clarity (Required)

For the first time F-actin is directly visualized in the Plasmodium parasite using ion beam milling and electron cryo-tomography, allowing for a 3D model of F-actin in the motile sporozoite. This is a major accomplishment in the field. Key new observations are: the presence of um long filaments, along the parasite length, the presence of large pores in the inner membrane complex at the parasite's basal end that were proposed to allow actin exchange from the pellicular space to the cytosol, and actin filament bundles in the nucleus.

****Minor comments:****

1. It is not correct that all in vitro studies observed only short actin filaments. Reference #6 showed de novo polymerized PfAct1 filaments as long as 30 um when visualized in real time by TIRF microscopy.

2. Elaborate briefly in the text regarding how the global F-actin concentration was obtained, or at least refer the reader to the Methods section where there are details.

3. I am not following the logic of why the authors state that actin filament disassembly is rate-limiting. Please elaborate.

4. Are the authors suggesting that the pellicular intermediate filaments they observe are analogous

to intermediate filaments found in mammalian cells? Is there any evidence for intermediate filament-like proteins in the Plasmodium parasite?

2. Significance:

Significance (Required)

This study is a major breakthrough because Plasmodium actin has not previously been directly visualized in the parasite. Plasmodium actin is much more dynamic and fragile than vertebrate actin and so the presence of long filaments was not necessarily expected. In addition, Plasmodium actin filaments cannot be visualized by standard methods such as fluorescent-phalloidin, although the use of the actin-chromobody has provided insights in related Apicomplexan parasites. Here the arrangement of unlabeled filaments and their length along the sporozoite was directly visualized for the first time. Novel features such as large pores in the IMC at the basal end of the parasite and the presence of nuclear actin were unexpected observations. The study was made possible by using state-of-the-art FIB-milling and electron cryo-tomography. Without doubt this study will make an impact in the Plasmodium field because the basis of gliding motility is the interaction of myosin motors with the visualized actin in the IMC. More broadly this study will be of interest to cell biologists interested in the actin cytoskeleton in higher organisms.

This initial study should also stimulate further research into the role of the Plasmodium nuclear actin, and the role of the limited repertoire of actin-binding proteins in assembly and disassembly of the actin filaments along the sporozoite length. The approach of FIB-milling and electron cryo-tomography may also allow a better description of the arrangement of the myosin motors, GAP and GAC proteins in the IMC with further study.

3. How much time do you estimate the authors will need to complete the suggested revisions:

Estimated time to Complete Revisions (Required)

(Decision Recommendation)

Less than 1 month

4. Review Commons values the work of reviewers and encourages them to get credit for their work. Select 'Yes' below to register your reviewing activity at Web of Science Reviewer Recognition Service (formerly Publons); note that the content of your review will not be visible on Web of Science.

Yes

Reviewer #1 (Evidence, reproducibility and clarity (Required)):

Pražák et al.: Molecular architecture of glideosome and nuclear F- actin in Plasmodium falciparum

The study by Pražák et al. uses cryo-electron microscopy to explore unresolved questions about the cellular and molecular architecture of the actin cytoskeleton in motile malaria parasite Plasmodium falciparum sporozoites (liver-infective forms). The key focus of the study is to determine the localization and organisation of actin filaments (F-actin). Employing a powerful combination of focused ion beam milling and electron cryo-tomography, the researchers visualize intact F-actin in sporozoites for the first time, detailing their organisation and potential journey, using this to define a model for their formation, progress and ultimate recycling within sporozoites (in relation to gliding motility/the sub-pellicular space) as well as observing actin in the nuclear periphery. The study reveals actin filaments that are longer than previously observed in vitro and how they concentrate organise within the cell in their "native" state.

Key Insights that are notable in the study:

- Visualization of F-actin: The authors present direct visualization of actin filaments in different phases of growth within Plasmodium falciparum cells, showing filaments much longer than previously seen in vitro (near micron in length rather than 100s of nm).
- Nuclear Actin Bundles: The authors observe extensive actin bundles within the nucleus, suggesting potential undefined roles.
- Pellicular Intermediate Filaments (PIFs): The authors identify a novel class of undefined filament - calling them PIFs - with no direct connection to subpellicular microtubules being observed.
- Actin Filament Dynamics: The authors present their data in the context of a well-accepted model for actin filament dynamics with assembly at the apical end, growth as they are transported down the cell, and disassembly at the basal end.
- Actin Exchange and Recycling: The authors observe large pores at the basal end of the IMC that they postulate these facilitate actin exchange between the pellicular space and cytosol, helping to maintain directional actin flow for efficient gliding motility.

Major comment:

- The study is beautifully executed and will clearly generate a lot of interest in the gliding/cytoskeletal field. I applaud the authors for that. My chief concern is, however, the numbers of cells involved AND whether the authors can be sure that their observations provide compelling evidence for the conclusions they make OR are the basis for limited hypothesis only. Most of my comments below lead directly from this concern about number of cells and a tendency (which I understand) to build hypotheses on the data observed. I feel at times the authors veer to close to speculation without drawing enough from the literature that has come before.

We thank the reviewer for their comments. We hope that by clarifying the much larger data set presented than understood by the reviewer (see below, ~85 sporozoites used in the paper rather than only 2) will clarify the basis on which our conclusions are made. Efforts have also been made to draw from a larger breadth of previous research in our discussions.

Questions:

1. "Subvolume averaging (SVA) was used to determine the structure and a complete 3D model of F-actin within sporozoites"

- To my reading, though I am uncertain, the entire paper is based on only 2 sporozoites? At least this is not qualified anywhere in the paper I can see easily (apologies if I have missed it). This needs clarification in the methodology and results sections.

- Assuming that few cells are the basis for observations, the conclusions made I would argue should be presented as speculative (e.g. based low numbers, n, of cells and individual observations). How many cells were used?

The data presented is derived from 29 tomograms containing ~85 individual sporozoites, which is comparable to other recent studies using focused ion beam milling and electron cryo-tomography (e.g.: in parasites: Gui, L., *et al.* (2023) Cryo-tomography reveals rigid-body motion and organization of apicomplexan invasion machinery. *Nat Commun*)).

This was not made sufficiently clear in our manuscript, stating only the number of tomograms used. We have added a sentence to clarify this:

"In total, volumes corresponding to ~85 individual cells from 29 tomograms were processed."

- Also - sporozoites in the salivary gland (the origin of these cells) are relatively immotile until they are released. Do the authors have data on whether the sporozoites were active or not when harvested? This will have implications for the state of F-actin observed and how it relates to gliding.

The sporozoites were released from salivary glands by mechanical crushing into Leibovitz medium with 10% heat inactivated human serum prior to shipping. This method has been thoroughly tested by TropiQ (where the sporozoites were isolated) and shown to result in infectious cells. Heat inactivated human serum has previously been shown to activate motility (Yang et al (2017) *Cellular microbiology*).

The method section has been clarified as below"

"Two weeks post infection, sporozoites were isolated using salivary gland dissection **and released using mechanical crushing** into Leibovitz medium supplemented with 10% heat inactivated human serum, prior to shipping at room temperature."

2. "a recently proposed model of severing mediated by coronin, an actin binding protein located at the basal end of gliding *P. berghei* sporozoites¹³".

- Coronin is not generally thought of as a disassembly factor (though it may be a minor player in this space). The ref cited (13, Bane et al.) demonstrates that following activation, coronin binds to actin filaments and then tracks with the protein to the rear end of the sporozoite. This fits its F-actin binding properties & membrane binding properties. I.e. it might just track engaged filaments. But my re-reading of the Bane paper does not suggest a model for coronin-dependent filament disassembly as far as I can see.

Please see reply to point after next below.

- What about membrane dynamics in general (plasma membrane retrograde flow <https://pubmed.ncbi.nlm.nih.gov/26792112/>)? This is largely absent from discussion in the paper.

The observations we can make with our technique prevent us from adding to discussion on membrane dynamics and retrograde flow but we agree it is an interesting and important point.

- Much more likely is ADF/cofilin playing a role in disassembly - though its generally cytoplasmic distribution (noted in *Toxoplasma* and *Plasmodium*) makes it hard to resolve exactly where it asserts its function.

While coronin is not considered as an actin filament severer directly, a mechanism of coronin-mediated filament severing has been shown in other eukaryotes. For this, coronin mediates a spatial-temporal recruitment of other filament regulators. In this mechanism, coronin binds first and, depending on the nucleotide state of the actin, then either shields the filament or allows access of ADFs to the filament (Gandhi et al (2009) PMID: 19450534; Galkin et al (2011) PMID: 22158895; Jansen et al (2015) PMID: 25995115; Mikati et al (2015) PMID: 26299936; Ge et al (2014) PMID: 25362487). Binding of these additional factors together results in filament destabilisation. In *Plasmodium*, overexpression of coronin, and not profilin or ADF2, rescued the pausing motility phenotype and salivary gland invasion of a *P. berghei* actin mutant sporozoite with stabilized actin filaments (Douglas et al (2018) PMID: 30011270; Yee et al (2022) PMID: 35998188). Together with the basal location of coronin in activated *P. berghei* sporozoites (original reference 13, Bane et al) and the observed shorter filaments at the basal end (this study), this makes for a reasonable case that such a mechanism is employed in this location of the sporozoite.

We have revised and expanded the sentence to make this point clearer, as follows:

“Short filaments at the basal pole supports a recently proposed model of severing mediated by actin binding protein coronin, likely through recruitment of actin depolymerizing factors (ADFs)¹⁴, as observed for other eukaryotes. In this mechanism, initial binding by coronin to the actin filament can either shield or allow access of ADFs, thereby enhancing severing and thus filament turnover¹⁵⁻¹⁹. The presence of coronin at the basal end of gliding *P. berghei* sporozoites²⁰, the observation that coronin overexpression rescued a gliding phenotype in an actin filament stabilized sporozoite^{8,14}, and the observation of shorter filaments at the basal end in comparison with middle parts of the sporozoite (this study), further suggests that such a mechanism could be employed in sporozoites.”

3. "at the basal pole, and in some cases form a one-filament deep shell, suggesting that actin disassembly may be rate-limiting".

- It's fine to speculate that shorter filaments at the basal pole is consistent with a model where this is the site of disassembly (no-brainer) but I don't understand the comment about rate limiting? What do the authors mean by this? The speed of disassembly limits the speed of the motile zoite? If so, what is the evidence that supports this beyond a simple accumulation of filaments here. That wasn't clear to me.

Thank you for this comment, we realise this was not phrased clearly enough to be understood as we intended. What we meant by this is that the rate of disassembly is clearly slower than the rate of filament accumulation at the basal end. This is interesting for a couple of reasons: 1) given that *in vitro*, *Plasmodium* actin disassembly rates are very high and 2) that this may indicate that this is one major rate limiting step in motility. We know from previous actin subdomain 4 mutants (which stabilises actin and lead to frequent motility pauses) that stabilisation has a consequence for turnover and thus for continuous motility.

We have now made this clearer in the text:

“The rate of filament disassembly is slower than the rate of filament accumulation at the basal pole, and in some cases filaments accumulate at the basal pole in a one-filament deep shell, suggesting that actin disassembly may be a rate-limiting step in motility (Fig. 1b,c). Previous results from actin filament stabilisation mutants show that filament

stabilisation has an influence on actin turnover and thus has consequences for continuous motility¹⁴.”

4. "The path of glideosome actin filaments towards the basal pole, may therefore be up an actin monomer gradient. The age of the filament together with this gradient could account for the elongation we observe as filaments move down the cell, prior to their disassembly at the basal pole. Localised pools of actin-stabilising proteins may also contribute to the length increase we observe."

- The missing piece of the puzzle here is nucleation. Where does it occur and does it continue down the length of the sporozoite. Formins (the primary nucleator in Plasmodium) concentrate at the sporozoite apex, but they may also track down the cell/IMC space. Since Formin's bind the barbed end (growth end) they can continue to grow filaments as they might migrate (if they stay with the filament). I feel the above sentence is predominantly speculation - the evidence from the imaging does not point to any model but supports data from other studies.

Our data as well as recent data from *Toxoplasma* and *Cryptosporidium* shows that the apical pole is the primary site of nucleation. In our data, we would expect to see a larger size range (more small filaments) at the sides of parasites if there were multiple sites of nucleation, rather than the clear increase in length that we observe. However, we agree that formins are important to mention and have adapted the sentence as below:

“The age of the filament **and action of formins**, together with this gradient could account for the elongation we observe as filaments move down the cell, prior to their disassembly at the basal pole.”

5. "We observed filamentous actin protruding through some basal pores, suggesting that filamentous actin can pass through into the cytoplasm... possible that these pores could facilitate more efficient exchange of G- and F-actin between the pellicular space and the rest of the cell."

- How many events is this observation based on? Are the authors sure that in the compendium of images by EM from the literature (from the likes of Bannister, Aikawa, Cyrklaff, Ferguson etc.) a) pores haven't already been described at the basal region of the IMC in plasmodium and b) that the odd structure consistent with an F-actin filament wasn't documented? This would make their argument far more convincing. If it's n=1/2 then it's quite speculative.

To our knowledge, Martinez et al., is the first published report of similar large and importantly, basal-restricted, IMC-spanning pores in apicomplexans, suggesting the same connection between these and the basal accumulation of actin.

The presence of large pores implies that there is transport of macromolecules taking place between the basal pellicular space and basal cytosol. The concentration of actin at the basal pellicular space is unusually high, increasing the probability that it is (one of) the species being transported. We have not observed any other filaments anywhere else in the dataset that could be misidentified as F-actin. Combined with the subvolume averaging data from the particular filament protruding through a basal pore, this gives a relatively high confidence in interpreting the density.

There is, of course, no way of knowing whether F-actin is routinely transported through these pores. This would only make sense for very short filaments that could be oriented perpendicularly to the IMC plane in the constrained pellicular space. The more plausible

hypothesis is that it is G-actin that diffuses through and our observation is (potentially) an unusual case where an existing filament in the cytosol “polymerised through” the pore.

This agrees with our interpretation of the data that “suggesting that filamentous actin can pass through”. We understand that this may have been misleading as it implies that it is specifically F-actin that passes through the pores.

We have therefore modified this section to clarify:

“The IMC in sporozoites has very few discontinuities, apart from at the basal pole which was dotted with ~25 nm diameter pores (Fig. 2b, S3). **These pores were observed almost exclusively at the basal end, leading us to hypothesise that their localization was related to the unusually high concentration of basal F-actin. Strikingly,** we observed filamentous actin protruding through or in direct proximity to basal pores (N=2), suggesting that filamentous actin can pass through ~~into the cytoplasm~~ (Fig. 2b, S3c, Videos 2,3). It is therefore possible that these pores could facilitate more efficient exchange of ~~G- and F-~~actin between the pellicular space and the rest of the cell. This would recycle actin and lower the concentration of actin at the rear of the sporozoite (up to 60 mM local F-actin concentration 100 nm from the basal pole end, **see methods**) and thus could allow for more efficient gliding. **The lack of similarly sized pores at other parts of the cell indicates** that their location at the basal pole is regulated, facilitating the directional flow of actin – into the pellicular space only via the pre-conoidal rings, and into the cytoplasm only via the basal pores.”

6. "lower the concentration of actin at the rear of the sporozoite (up to 60 mM local F-actin concentration 100 nm from the basal pole end) and thus could allow for more efficient gliding"

- Where does this number 60 come from?

Methods have been expanded to include the determination of local actin concentration, as below, and reference made to this in the main text.

“**For local actin concentration, the volume of the pellicular space was measured using manually segmented models at the basal end of a sporozoite oriented roughly perpendicularly to the FIB-sectioning axis. (sporozoite 1, Figure 1c, h). 8.73×10^{-20} moles of actin (based on subvolume averaging data) in 1.53×10^{-18} L of pellicular space corresponds to a local concentration of $5.7 \times 10^{-2} \text{ molL}^{-1} \sim 60 \text{ mM}$.**”

7. "In two cellular locations we observed glideosomal actin filaments bound by densities consistent with myosin heads, with tails leading to PIFs in the IMC".

- I would love this to be the case (many in the field have long sought to visualise the motor). However, if this is simply based on n=2 densities (in 1 cell?) it feels very speculative.

We are confident that the two densities that we observe are myosins. This is simply an observation (which we think is different from a speculation). We have observed dozens of potential myosins but only two that we feel confident in reporting and believe that by clearly stating that N=2, we are clearly stating that this is an observation that readers can make their own conclusions from, and that we would definitely like to follow up in the future publication. However, to clarify this further, we have added a sentence, as below:

“In two cellular locations we observed glideosomal actin filaments bound by densities consistent with myosin heads, with tails leading to TPFs in the IMC. **This suggests that the tail is anchored in the IMC via TPFs, however this will need to be validated in the future.**”

8. "While there have been many reports of the presence of nuclear actin in different organisms, this is to our knowledge, the first direct evidence of nuclear F-actin in the

absence of staining or stabilising agents in any organism."

- The caveat at the end of this sentence is a bit unreasonable to my mind. Back in 2008 Cryo-SEM showed F-actin in the nuclear periphery of *Xenopus* oocytes (<https://pubmed.ncbi.nlm.nih.gov/19017237/>). It is true this requires some level of shadowing etc. but is that labelling? Does it matter? There is a whole field of research exploring nuclear actin across eukaryotes which feels a bit overlooked here.

We make this point precisely because there is a whole field of nuclear actin research. However, the existence and function of nuclear F-actin in unperturbed cells is still a matter of debate (e.g. <https://www.nature.com/articles/nrm3681>). A common method of nuclear F-actin visualisation is by using fluorescent phalloidin, a known F-actin stabilising compound. Others (notably the cited study of *Xenopus* oocytes) make use of fixation followed by staining, which again is known to induce artefacts. There are studies making use of freeze-substitution to avoid fixation artefacts, but these again rely on indirect labelling. In other words these show filaments, which very likely consist of actin.

Prior to our work, a number of studies have been published showing cryoET data from FIB milled nuclei and none have reported the presence of actin filaments. Therefore drawing attention to this being the first direct observation seemed significant.

9. "Whether this simply serves as an intra-nuclear protective structure or fulfils additional molecular roles, such as mechanosensing, in the nucleus requires further investigation."

- Actin has been implicated in numerous functions in Apicomplexa and the authors would do well to cite much more of the literature, otherwise there is a sense that this is entirely uncharted territory. The work of the Scherf lab (e.g. <https://pubmed.ncbi.nlm.nih.gov/22100161/>), Cowman lab (e.g. <https://pubmed.ncbi.nlm.nih.gov/22264509/>) and my own lab (e.g. <https://pubmed.ncbi.nlm.nih.gov/22389687/>) all discuss potential roles for nuclear actin that are worth considering in the context of having imaged it with new resolution.

We have now added more discussion and references on the previously identified roles of actin in parasite nuclear function. Of course, as indicated in the discussion, whether such roles might be played in the gliding sporozoite, or whether it is more structural/protective will require further research.

"Indeed, actin signals have indicated F-actin accumulation near to the nucleus (in ~20% of sporozoites **in the case of the actin chromobody**) during motility and invasion of apicomplexan parasites, suggesting that a nuclear actin cage facilitates efficient invasion and/or protects the nucleus from damage when the parasite undergoes constriction^{8,9,23}. However, here we have observed extensive bundles within the nucleus itself (Fig. 1a and 2d). ***Plasmodium* actin has previously been implicated in spatial repositioning of genes and histone methylation states in monoallelic expression of blood stages^{24,25}**. Whether these actin bundles observed in sporozoites serve as an intra-nuclear protective or mechanosensing structure, or fulfil additional molecular roles **in gene expression** in the nucleus requires further investigation."

Minor Comment:

In the supplementary movies, could the authors not include the (now standard) cryo-EM fly through where the sections are transitioned through with addition of rendering to build the 3D models shown in the body of the paper. This would give a much clearer interpretation of where F-actin is.

Together with the original 4 videos where we annotate densities in our raw tomograms, we have now added 4 extra videos showing the raw tomograms and merging through to our 3D segmentations. We hope these help with data interpretation.

Reviewer #1 (Significance (Required)):

- There is the nugget of something very important in this paper. My chief concern is the limitations that would come from the numbers of cells imaged (ensuring the data is properly quantified), and whether this is substantial enough to then interpret and build robust hypotheses from.
- With robust data (i.e. many cells, perhaps in different states of motility/non-motility) this paper would then become the foundation for others to go and test (at molecular and cellular levels) whether the hypotheses raised are correct and ultimately resolve the key question - what is F-actin doing in the sporozoite and how it is regulated?
- The cell biology/parasitology community would be very interested in such a paper!
- I have had a long-standing interest in the cell biology of apicomplexan parasites (plasmodium specifically) and worked for many years on the actomyosin gliding motor.

We thank the reviewer for their thorough reading and probing of our work. Hopefully, this concern will have been resolved with the more clearly reported number of cells studied (~85 individual sporozoites). We are also excited by the possibility of investigating the molecular architecture of actin at different motility states of the parasite, but we believe this would merit a separate publication.

Reviewer #2 (Evidence, reproducibility and clarity (Required)):

For the first time F-actin is directly visualized in the Plasmodium parasite using ion beam milling and electron cryo-tomography, allowing for a 3D model of F-actin in the motile sporozoite. This is a major accomplishment in the field. Key new observations are: the presence of μm long filaments, along the parasite length, the presence of large pores in the inner membrane complex at the parasite's basal end that were proposed to allow actin exchange from the pellicular space to the cytosol, and actin filament bundles in the nucleus.

Minor comments:

1) It is not correct that all in vitro studies observed only short actin filaments. Reference #6 showed de novo polymerized PfAct1 filaments as long as 30 μm when visualized in real time by TIRF microscopy.

Thank you for highlighting this. We have changed the sentence to the below:

“However, *Plasmodium* F-actin appears to be dynamically unstable *in vitro* with very high disassembly and fragmentation rates⁴⁻⁷.”

2) Elaborate briefly in the text regarding how the global F-actin concentration was obtained, or at least refer the reader to the Methods section where there are details.

Thank you for this comment, we have now referenced the method section in our main text.

“The global F-actin concentration was measured to be $40 \pm 7 \mu\text{M}$ (see Methods).”

3) I am not following the logic of why the authors state that actin filament disassembly is rate-limiting. Please elaborate.

Thank you, we realise that this may have not been as clear as we intended as reviewer 1 also pointed this out. What we meant by this is that the rate of actin filament disassembly is slower than the rate of filament accumulation at the basal end. Please see our discussion in response to this from reviewer one and note that we have changed the text to say:

“The **rate of filament disassembly is slower than the rate of filament accumulation** at the basal pole, and in some cases **filaments accumulate at the basal pole in** a one-filament deep shell, suggesting that actin disassembly may be a rate-limiting step **in motility** (Fig. 1b,c). **Previous results from actin filament stabilisation mutants show that filament stabilisation has an influence on actin turnover and thus has consequences for continuous motility¹⁴.**”

4) Are the authors suggesting that the pellicular intermediate filaments they observe are analogous to intermediate filaments found in mammalian cells? Is there any evidence for intermediate filament-like proteins in the Plasmodium parasite?

No, we are not suggesting this and agree this naming may be confusing. To avoid confusion, we have now changed the name to TPFs – thin pellicular filaments.

Reviewer #2 (Significance (Required)):

Significance

This study is a major breakthrough because Plasmodium actin has not previously been directly visualized in the parasite. Plasmodium actin is much more dynamic and fragile than vertebrate actin and so the presence of long filaments was not necessarily expected. In addition, Plasmodium actin filaments cannot be visualized by standard methods such as fluorescent-phalloidin, although the use of the actin-chromobody has provided insights in related Apicomplexan parasites. Here the arrangement of unlabeled filaments and their length along the sporozoite was directly visualized for the first time. Novel features such as large pores in the IMC at the basal end of the parasite and the presence of nuclear actin were unexpected observations. The study was made possible by using state-of-the-art FIB-milling and electron cryo-tomography. Without doubt this study will make an impact in the Plasmodium field because the basis of gliding motility is the interaction of myosin motors with the visualized actin in the IMC. More broadly this study will be of interest to cell biologists interested in the actin cytoskeleton in higher organisms.

This initial study should also stimulate further research into the role of the Plasmodium nuclear actin, and the role of the limited repertoire of actin-binding proteins in assembly and disassembly of the actin filaments along the sporozoite length. The approach of FIB-milling and electron cryo-tomography may also allow a better description of the arrangement of the myosin motors, GAP and GAC proteins in the IMC with further study.

We thank the reviewer for their comments. We are excited by these findings and hope that they will be of use to the community in stimulating new research avenues.

Dear Dr. Ferreira,

Thank you for submitting your revised manuscript, which was previously peer reviewed at Review Commons. It has now been seen by one of the original referees.

As you can see, both referees find that the study is significantly improved during revision and recommend publication. However, I need you to address the points below before I can accept the manuscript.

- We believe that the format of the manuscript is better suited for our Reports format rather than Scientific Article, which needs to be updated accordingly during the resubmission. Please make an Introduction section and a combined Results & Discussion as per our format requirements. In Please see <https://www.embopress.org/page/journal/14693178/authorguide#researcharticleguide>
- Please fill out and include an author checklist as listed in our online guidelines (<https://www.embopress.org/page/journal/14693178/authorguide>)
- Main figures should be removed from the manuscript text and uploaded as individual, high resolution figure files. The legends should remain in the manuscript text and be moved after the References. Supplemental figures should be renamed "Figure EV1" etc. and also uploaded as individual, high resolution figure files. The legends should be moved to the manuscript text, after the main figure legends, under the heading "Expanded View Figure Legends"
- Please provide 3-5 keywords for your study. These will be visible in the html version of the paper and on PubMed and will help increase the discoverability of your work.
- Please remove the following statement from the Data Availability section "All data are available on request." Also, please make the dataset EMD- 19898 publicly available and provide a link which directly resolves to the dataset. Please see <https://www.embopress.org/page/journal/14693178/authorguide#dataavailability> for further information.
- We note the following regarding the funding information. The Hessian Excellence Program and EU's Horizon 2020 programme (grant agreement No 731060) is currently missing from the manuscript tracking system. Wellcome Trust 209250/Z/17/ Z and 107806/Z/15/Z are mentioned in the manuscript tracking system, but not in the Acknowledgements section of the manuscript.
- Please remove the Author Contributions section from the manuscript.
- Please change the title of Competing interests section to Disclosure Statement and Competing Interests.
- As per our format requirements, in the reference list, citations should be listed in alphabetical order and then chronologically, with the authors' surnames and initials inverted; where there are more than 10 authors on a paper, 10 will be listed, followed by 'et al.'. Please see <https://www.embopress.org/page/journal/14693178/authorguide#referencesformat>
- The legends for the videos should be removed from the suppl. materials file and each legend should be zipped with the corresponding video file. The videos should be renamed "Movie EV1" - EV7.
- Please submit source data as requested by our source data coordinator Dr. Hannah Sonntag.
- All research articles submitted as revised versions must include a structured methods section that includes a Reagents and Tools Table followed by a Methods and Protocols section. Please see <https://www.embopress.org/page/journal/14693178/authorguide#structuredmethods> for further information.
- All panels for the main figures need to be called out and in sequential order, i.e. Fig 1A-G before Fig 2, etc.
- The heading "Abstract" is missing.
- Order of the manuscript sections should be as follows: Abstract, Keywords, Introduction, Results, Discussion, Methods, Acknowledgements, Disclosure and competing interests statement, References, Figure legends, Tables and their legends, Expanded View Figure legends
- We note a potential reuse of images in Figure 2E and Figure 3C, please clarify. If reuse is essential, please clarify in legends of both figures.
- Please update the citation to the preprint (34) as follows: Citations to manuscripts posted on recognized preprint servers can be cited the following way:
In-text citation: (preprint: NAME1 et al, YEAR)
Reference list: Author NAME1, Author NAME2, (YEAR) article title. bioRxiv doi: 1234/002.dj123 [PREPRINT]
- Our production/data editors have asked you to clarify several points in the figure legends:
 - o Please indicate the statistical test used for data analysis in the legend of figure 3D.
- Papers published in EMBO Reports include a 'synopsis' and 'bullet points' to further enhance discoverability. Both are displayed on the html version of the paper and are freely accessible to all readers. The synopsis includes a short standfirst summarizing the study in 1 or 2 sentences (max 35 words) that summarize the paper and are provided by the authors and streamlined by the handling editor. I would therefore ask you to include your synopsis blurb and 3-5 bullet points listing the key experimental findings.
- In addition, please provide an image for the synopsis. This image should provide a rapid overview of the question addressed in the study but still needs to be kept fairly modest since the image size cannot exceed 550 (width) x 300-600 (height) pixels.

Thank you again for giving us to consider your manuscript for EMBO Reports, I look forward to your minor revision.

Kind regards,

Deniz Senyilmaz Tiebe

--

Deniz Senyilmaz Tiebe, PhD
Senior Scientific Editor
EMBO Reports

Referee #1:

The clarification made and the substantial textual changes in the revised manuscript address all of my concerns. I believe the changes and the substantial supplementary movies plus extra clarity have really transformed the paper from what felt speculative to what is now robust and sound science. If my original review was a little skeptical, my opinion now is entirely positive. This is an excellent study and the authors should be commended on a solid and very insightful body of work. There is no doubt in my mind this work will generate a lot of enthusiasm in the cytoskeletal and gliding motility fields (and of course the ever growing CryoEM world).

I would support publication without need for further revision.

All editorial and formatting issues were resolved by the authors.

Dr. Josie Ferreira
University College London
Structural and Molecular Biology
London
United Kingdom

Dear Josie,

Thank you for submitting your revised manuscript. I have now looked at everything and all is fine. Therefore, I am very pleased to accept your manuscript for publication in EMBO Reports.

Congratulations on a nice work!

Kind regards,

Deniz
--
Deniz Senyilmaz Tiebe, PhD
Senior Scientific Editor
EMBO Reports
Your manuscript will be processed for publication by EMBO Press. It will be copy edited and you will receive page proofs prior to publication. Please note that you will be contacted by Springer Nature Author Services to complete licensing and payment information.

Rev_Com_number: RC-2024-02609
New_manu_number: EMBOR-2024-60683V2
Corr_author: Ferreira
Title: Molecular architecture of glideosome and nuclear F-actin in Plasmodium falciparum